# A Comparative Study of the Impact of NO-Related Agents on MK-801- or Scopolamine-Induced Cognitive Impairments in the Morris Water Maze

**DOI:** 10.3390/brainsci13030410

**Published:** 2023-02-27

**Authors:** Paulina Cieślik, Magdalena Borska, Joanna Monika Wierońska

**Affiliations:** Department of Neurobiology, Maj Institute of Pharmacology Polish Academy of Sciences, Smętna 12, 31-343 Kraków, Poland

**Keywords:** Morris water maze, spatial memory, DETA NONOate, spermine NONOate, nNOS inhibitor

## Abstract

Learning and memory deficits accompany numerous brain dysfunctions, including schizophrenia and Alzheimer’s disease (AD), and many studies point to the role of nitric oxide (NO) in these processes. The present investigations constitute the follow-up of our previous research, in which we investigated the activity of NO releasers and a selective inhibitor of neuronal NO synthase (nNOS) to prevent short-term memory deficits in novel object recognition and T-maze. Here, the ability of the compounds to prevent the induction of long-term memory deficits by MK-801 or scopolamine administration was investigated. The Morris Water Maze test, a reliable and valid test of spatial learning and memory, was used, in which escape latency in the acquisition phase and nine different parameters in the retention phase were measured. A fast NO releaser (spermine NONOate), a slow NO releaser (DETA NONOate), and a nNOS inhibitor, N(ω)-propyl-L-arginine (NPLA), were used. The compounds were administered i.p. at a dose range of 0.05–0.5 mg/kg. All compounds prevented learning deficits in the acquisition phase and reversed reference memory deficits in the retention phase of the scopolamine-treated mice. Spermine NONOate was the least effective. In contrast, the drugs poorly antagonised MK-801-induced deficits, and only the administration of DETA NONOate induced some improvements in the retention trial.

## 1. Introduction

Nitric oxide (NO) is a critical molecule implicated in a wide range of functions and pathologies with pleiotropic effects in the central nervous system (CNS). The molecule is synthesized from l-arginine in the presence of NADPH by constitutive forms of nitric oxide synthases (NOS), endothelial (eNOS), neuronal (nNOS), or inducible NO synthase (iNOS). Endothelial NOS is primarily responsible for the generation of NO in the vascular endothelium, a monolayer of flat cells lining the interior surface of blood vessels at the interface between circulating blood in the lumen and the remainder of the vessel wall. NO produced by eNOS in the vascular endothelium plays a crucial role in processes such as regulating vascular tone or cellular proliferation. Therefore, functional eNOS is essential for a healthy vascular system, which, e.g., ensures proper functioning of the CNS [1,2].

Neuronal NOS generates NO in neurons, which may exist in particulate and soluble forms at differential subcellular localizations. NO cannot be stored in cells, and it depends on continual synthesis to exert its functional properties [3]. 

Upon activation of N-methyl-D-aspartate (NMDA) receptors, nNOS generates a neuronal pool of NO that exerts physiological effects by changing cGMP levels, forming the so-called glutamate-NO-cGMP pathway, which has been shown to be important in the mechanisms that regulate learning and memory processes [4,5]. Inhibition of nNOS interferes with the learning of some spatial tasks [6,7], indicating that increased and not decreased nNOS-dependent NO activity is critical in cognition. Furthermore, diminished activity of the glutamate-NO-cGMP pathway and subsequent decreased levels of cGMP result in cognitive impairment [4]. However, contradictory research indicates that increased nNOS activity and expression may contribute to the pathophysiology of many CNS diseases, and hence, inhibiting nNOS could exert therapeutic effects. A variety of studies have demonstrated that NO acts as a “double edged sword” and that both its activation and inhibition may contribute to pathological or therapeutic effects to the same extent. Similar results concerning the activities of a fast NO releaser, spermine NONOate (t = 30 min), a slow NO releaser, DETA NONOate (t = 20 h), and a highly selective nNOS inhibitor, N(ω)-propyl-L-arginine (NPLA), have also been published recently by our group [8]. Previous studies were performed on the tests evaluating short-term memory deficits (novel object recognition, Y-maze).

The present study is a continuation of our previous investigations. Here, the activity of the compounds was evaluated in the Morris water maze, a reliable and valid test of spatial learning and memory [9,10]. Learning deficits were induced by scopolamine (a non-selective muscarinic receptor antagonist) or MK-801 (a NMDA receptor antagonist), regarded as experimental models of AD [11,12] or schizophrenia [13,14], respectively. 

NMDA or muscarinic receptors are expressed on neurons, endothelial cells, and astrocytes, which are closely related to each other and create the minimally functioning unit in the brain called the neurovascular unit (NVU) [15]. Nitric oxide-dependent pathways are essential regulators of mutual interactions between cells, and the fluctuations of NO-dependent activities (such as cGMP synthesis or the production of oxidative stress) translate into NVU dysfunctions that may contribute to the amnestic effect of MK-801 or scopolamine and at least partially resemble those observed in schizophrenic or AD patients. We thus hypothesise that the administration of NO-related agents may counteract the MK-801- or scopolamine-induced dysfunctions by restoring brain homeostasis. 

## 2. Materials and Methods

### 2.1. Animals

Male CD-1 mice were used in the study (Charles River, Germany). The animals were kept under standard conditions (12:12 light-dark cycle, 22 ± 2 °C, and 55 ± 10% humidity), with water and food available ad libitum. Behavioural procedures were performed during the light phase of the cycle. All drugs were administered intraperitoneally (i.p.) at a volume of 10 mL/kg. The procedures were conducted in accordance with the European Communities Council Directive of 22 September 2010 (2010/63/EU) and Polish legislation acts concerning animal experimentation, and they were approved by the II Local Ethics Committee at the Maj Institute of Pharmacology, Polish Academy of Sciences in Krakow.

### 2.2. Drugs

MK-801 (Hello Bio Inc., 304 Wall Street, Princeton, NJ 08540, USA) and scopolamine (Tocris Bioscience/Biotechne Corporation, Minneapolis, MN, USA) were dissolved in 0.9% saline. Spermine NONOate and DETA NONOate (Abcam, Cambridge, UK) were dissolved in 0.9% saline, and NPLA was dissolved in a small amount of DMSO and then adjusted to the proper volume with 0.9% saline. When the administration of experimental compounds was omitted (control, MK-801, or scopolamine group), the animals received appropriate vehicles. The doses used in behavioural experiments were based on our previous studies [8]. 

### 2.3. Morris Water Maze Protocol

Different and contradictory results regarding the activity of NO donors and inhibitors have been published in previous studies using various protocols and animal species. Thus, in the present studies, to avoid inconsistency in experimental protocols, the experiments were performed in comparable sets of studies under the same conditions and on one species of animal (CD1 mice).

The test was performed according to Sałat et al. [16,17] with minor modifications. Mice were trained to find a hidden platform (10 cm in diameter, approximately 1 cm beneath the water surface) in a circular pool (120 cm in diameter) using spatial cues. The time to reach the platform in the acquisition trial has been defined as a measurement of spatial learning. Visual cues were consistent during the experiments and included geometric figures placed in the proximity of the maze, the experimenters, and differently coloured walls surrounding the maze. The water temperature was maintained at 22–23 °C. The pool was divided into 4 quadrants: NE, NW, SE, and SW, and the platform was located in the NE quadrant (target zone TZ) (Figure 1). ANY-MAZE (Stoelting Europe, Dubin, Ireland) was used to track the animals during the MWM test.

The acquisition phase consisted of 4 (scopolamine experiments) or 5 training days (MK-801 experiments), with 4 training trials per day (the difference in the duration of the acquisition phase was due to the differences in the drugs’ activities, as explained below in the discussion section). During each trial, the animal was placed gently in the pool and was allowed to explore it for up to 60 s to find the hidden platform (if the platform was found earlier, the animal was removed from the platform and placed in the home cage). Starting locations (1–4, Figure 1) were selected in a pseudorandom manner for each training day. The inter-trial interval was 10 min.

Spermine NONOate, DETA NONOate, or NPLA were administered 30 min before MK-801 or scopolamine, which were administered 30 min before the first training trial each day. The administration schedule is presented in Figure 2.

The retention trial was performed on the day after the acquisition phase. Before the test, the platform was removed from the pool, and no drugs were administered. Mice were released from the WS start point and were allowed to swim freely for 60 s.

### 2.4. Analysis

With the use of the ANY-MAZE program, the following parameters were analyzed. During the acquisition phase, escape latency (in seconds), defined as the time required for animals to reach the hidden platform (means ± SEM are shown for each group on each training day), was measured. During the 60 s test (retention trial), several parameters affected by memory-disruptive agents were measured, grouped as target zone (TZ)-related parameters (latency to reach TZ, time oriented towards the centre of the TZ, time spent in the TZ, distance travelled until 1st entry to the TZ, distance travelled in the TZ, and number of entries into TZ) and platform zone (PZ)-related parameters (latency to reach the PZ, number of PZ crossings, time in the PZ, and distance travelled until first entry into the PZ). 

For each treatment, thermal charts were analysed which visualise the duration of animal swimming in specific areas of the maze. 

### 2.5. Statistics

Statistical analyses were performed as follows:(a)Repeated-measures ANOVA was used to evaluate escape latency (the measurement of spatial learning) between groups during the training days in the acquisition phase;(b)The Student’s t-test was used to compare controls vs. MK-801 or controls vs. scopolamine-treated animals (which was applied to all tested parameters in the retention phase);(c)A one-way ANOVA followed by a Tukey’s post hoc comparison test was used to evaluate the activity of the compounds in the retention phase (applied to all tested parameters). Treatment groups receiving drugs + MK-801 were compared with the MK-801 group, and treatment groups receiving drugs + scopolamine were compared with the scopolamine group.

The data were analysed using GraphPad Prism (version 9.4.1) and are presented as the means ± SEM.

## 3. Results

### 3.1. Acquisition Phase

Administration of scopolamine or MK-801 impaired the ability of the mice to learn the location of the hidden platform (Figure 3 and Figure 4).

Spermine NONOate at all tested doses [F_(3.36)_ = 10.268; *p* < 0.05)] and NPLA at the doses of 0.05 and 0.1 mg/kg (F_(3.36)_ = 5.61; *p* < 0.01) reversed scopolamine-induced disruptions (Figure 3) in the acquisition phase.

None of the tested compounds reversed the MK-801-induced impairments in the acquisition phase (Figure 4).

When administered alone, the investigated compounds had no impact on escape latencies during the acquisition phase. The effects of spermine NONOate, DETA NONOate, and NPLA administered at doses of 0.5 mg/kg were comparable to the controls (data not shown).

### 3.2. Retention Trial

#### 3.2.1. Target Zone—Related Parameters

When calculated with the Student’s t-test (compared to control animals) scopolamine administration impaired all six TZ-related parameters tested: latency to the first entry into the TZ (t = 3.54; df = 42; *p* < 0.001), time in the TZ (t = 3.23; df = 43; *p* < 0.002), distance travelled in the TZ (t = 3.84; df = 43; *p* < 0.0004), distance travelled until first entry into the TZ (t = 2.82; df = 43; *p* < 0.007), number of entries into the TZ (t = 4.26; df = 41; *p* < 0.0001), and the time oriented towards the centre of the TZ when inside the zone (t = 2.704; df = 43; *p* < 0.009) (Figure 5, Figure 6 and Figure 7). A one-way ANOVA followed by a Tukey’s post hoc comparison test revealed that spermine NONOate reversed the effect of scopolamine on latency to first entry into the TZ at the doses of 0.1 and 0.5 mg/kg [F_(3.32)_ = 3.8; *p* = 0.01] and reversed the scopolamine-induced increased distance travelled until first entry into the TZ zone [F_(3.32)_ = 5.05; *p* = 0.005] (Figure 5). No activity regarding the time in the TZ [F_(3.32)_ = 0.91, *p* = 0.44], the distance travelled in the TZ [F_(3.32)_ = 1.55, *p* = 0.21], the number of entries into the TZ [F_(3.32)_ = 2.11, *p* = 0.11], or the time oriented towards the centre of the TZ [F_(3.32)_ = 0.2, *p* = 0.87] was observed (Figure 6).

DETA NONOate mitigated the impact of scopolamine on latency to first entry into the TZ at doses of 0.1 and 0.5 mg/kg [F_(3.30)_ = 6.89; *p* = 0.001], time in the TZ at all investigated doses [F_(3.31)_ = 3.92; *p* = 0.01], distance travelled in the TZ at doses of 0.05 and 0.1 mg/kg [F_(3.30)_ = 3.83; *p* = 0.01], and distance travelled until first entry into the TZ at doses of 0.1 and 0.5 mg/kg [F_(3.31)_ = 4.38; *p* = 0.01]. There is no activity regarding the number of entries into the TZ [F_(3.31)_ = 1.4, *p* = 0.25] or the time oriented towards the centre of the TZ when inside the zone [F_(3.31)_ = 1.76, *p* = 0.17].

NPLA reversed scopolamine-induced deficits in all the measured parameters: latency to the first entry into the TZ at the doses of 0.05, 0.1, and 0.5 mg/kg [F_(3.33)_ = 5.76; *p* = 0.003], time in the TZ at the dose of 0.05 mg/kg [F_(3.33)_ = 2.89; *p* = 0.05], distance travelled until first entry into the TZ at the doses of 0.05, 0.1, and 0.5 mg/kg [F_(3.33)_ = 7.0; *p* = 0.0009], distance travelled in the TZ at the dose of 0.1 mg/kg [F_(3.33)_ = 3.0; *p* = 0.04], time oriented towards the centre of the TZ when inside the zone at the dose of 0.1 mg/kg [F_(3.33)_ = 8.48; *p* = 0.0003], and number of entries into the TZ at the dose of 0.05 mg/kg [F_(3.33)_ = 3.1; *p* = 0.03] (Figure 7).

The administration of MK-801 impaired the latency to first entry into the TZ (t = 2.43; df = 46; *p* < 0.01), the time spent in the TZ (t = 3.94; df = 47; *p* < 0.0003), the distance travelled in the TZ (t = 4.26; df = 47; *p* < 0.0001), the number of entries into the TZ (t = 4.63; df = 47; *p* < 0.0001), and the time oriented towards the centre of the TZ when inside the zone (t = 2.04; df = 47; *p* < 0.04) (Figure 8, Figure 9 and Figure 10). 

Spermine NONOate reversed the effect of MK-801 on the latency to first entry into the TZ at a dose of 0.5 mg/kg [F_(3.31)_ = 3.9; *p* = 0.01] (Figure 8). No activity regarding the time in the TZ [F_(3.31)_ = 1.14, *p* = 0.34], the distance travelled in the TZ [F_(3.31)_ = 1.35, *p* = 0.27], the number of entries into the TZ [F_(3.31)_ = 3.1, *p* = 0.03], or the time oriented towards the centre of the TZ when inside the zone [F_(3.31)_ = 1.13, *p* = 0.35] was observed (Figure 8).

DETA NONOate reversed the latency to the first entry into the TZ at the dose of 0.05 mg/kg [F_(3.32)_ = 1.94; *p* < 0.1], the time in the TZ at the dose of 0.5 mg/kg [F_(3.32)_ = 3.35, *p* = 0.03], and the time oriented towards the centre of the TZ at the doses of 0.05 and 0.5 mg/kg [F_(3.32)_ = 4.09; *p* = 0.01]. No activity regarding the distance travelled in the TZ [F_(3.32)_ = 2.32, *p* = 0.09] or the number of entries into the TZ [F_(3.32)_ = 1.29, *p* = 0.29] was observed (Figure 9).

Reversed MK-801-induced NPLA increased the latency to first entry into the TZ at a dose of 0.5 mg/kg [F_(3.28)_ = 3.41; *p* = 0.03]. (Figure 10). No activity regarding the time in the TZ [F_(3.28)_ = 1.44, *p* = 0.25], the distance travelled in the TZ [F_(3.28)_ = 0.94, *p* = 0.4], the number of entries into the TZ [F_(3.28)_ = 1.59, *p* = 0.2], or the time oriented towards the centre of the TZ when inside the zone [F_(3.28)_ = 0.46, *p* = 0.7] was observed (Figure 10).

#### 3.2.2. Platform Zone-Related Parameters

Analysis concerning the impact of scopolamine on PZ-related parameters revealed deteriorations in the latency to first entry into the PZ (t = 2.47; df = 35; *p* < 0.006), the time spent in the PZ (t = 4.38; df = 43; *p* < 0.0001), the distance travelled until first entry into the PZ (t = 2.47; df = 35; *p* < 0.01), and the number of entries into the PZ (t = 4.38; df = 43; *p* < 0.0001) (Figure 11, Figure 12 and Figure 13). 

Spermine NONOate reversed scopolamine-induced disruptions in the latency to first entry into the PZ at the dose of 0.05 mg/kg [F_(3.24)_ = 3.33; *p* < 0.03] and the distance travelled until first entry into the PZ at the dose of 0.05 mg/kg [F_(3.24)_ = 6.46; *p* < 0.002] (Figure 11). No activity regarding the time in PZ [F_(3.24)_ = 1.57, *p* = 0.2] or the number of entries into the PZ [F_(3.26)_ = 1.32, *p* = 0.29] was observed.

The administration of DETA NONOate to some extent reversed scopolamine-induced deficits in latency to the first entry to the PZ at a dose of 0.5 mg/kg [F_(3.23)_ = 2.28; *p* = 0.05] and distance travelled until the first entry into the PZ at the dose of 0.5 mg/kg, [F_(3.23)_ = 2.22, *p* = 0.11] (however, statistical significance was not achieved). No activity regarding the time in the PZ [F_(3.22)_ = 0.94, *p* = 0.4] or the number of entries into the PZ [F_(3.22)_ = 0.67, *p* = 0.57] was observed (Figure 12). 

NPLA was not effective at either dose in any of the parameters tested (Figure 13). Latency to the first entry [F_(3.23)_ = 0.46, *p* = 0.71], the time in the PZ [F_(3.23)_ = 1.41, *p* = 0.25], the distance travelled until the first entry into the PZ [F_(3.23)_ = 0.5, *p* = 0.68], and the number of entries into the PZ [F_(3.23)_ = 2.3, *p* = 0.1].

The impact of MK-801 administration was observed in the number of entries into the PZ (t = 2.86; df = 47; *p* < 0.006) and the time spent in the PZ (t = 2.2; df = 47; *p* < 0.03). 

Analysis of the MK-801-induced behaviours revealed no activity of the selected compounds on PZ-related activities (Figure 14). Spermine NONOate: number of entries into the PZ [F_(3.28)_ = 1.59, *p* = 0.2], the time in the PZ [F_(3.28)_ = 0.46, *p* = 0.7]; DETA NONOate: the number of entries into the PZ [F_(3.28)_ = 0.55, *p* = 0.64], the time in the PZ [F_(3.28)_ = 1.68, *p* = 0.19]; and NPLA: the number of entries into the PZ [F_(3.28)_ = 1.32, *p* = 0.28], the time in the PZ [F_(3.28)_ = 1.05, *p* = 0.36]. 

When administered alone, the compounds had no impact on memory retention (Table 1).

Heat maps reflecting the time that mice spent in particular areas of the maze show the animal’s search strategies (Figure 15).

## 4. Discussion

In the present study, we investigated the effect of the NO releasers (DETA NONOate and spermine NONOate) and a nNOS inhibitor (N(ω)-propyl-L-arginine, NPLA) on learning performance in the Morris water maze test, which is widely used in rodents to investigate spatial memory and learning. Its basic principle is that the animal is placed in a large circular pool, and using various cues, it is required to find a platform that allows it to escape the water [18,19]. A secondary aim of the work was to compare the amnestic effects of scopolamine and MK-801. To the best of our knowledge, this is the first complex comparative study concerning the impact of scopolamine and MK-801 on Morris water maze spatial learning in CD1 mice. 

Escape latency, defined as the time needed by the animal to find the platform, was the basic parameter measured each day of training and was regarded as a measurement of spatial learning, which was impaired by scopolamine and MK-801. The impact of both compounds was comparable to other studies [16,17,20,21,22]. On the test day, the escape platform was removed, and the mice were allowed to search for it for a fixed time (60 s). Detailed analysis of the parameters affected in the MWM was performed and included not only the latency to the first entry into the TZ or PZ but also behaviours focused on active searching for a place where the hidden platform was located. The parameters related to PZ reflect spatial learning and memory better, as the animals are required to navigate and locate a relatively small area with the use of only a few cues. This constitutes a big challenge, and any randomness is minimized. The possible advantages of the measurements of different parameters were reviewed elsewhere by Maei et al. [23]. 

Our results indicate that the preventive effect of the compounds on scopolamine-induced impairments in spatial learning during the acquisition phase does not necessarily reflect their impact on the reference memory measured in the retention trial. The efficacy of DETA NONOate during the acquisition phase was weaker than spermine NONOate or NPLA, but DETA NONOate-treated animals performed best in the retention trial. 

Due to the lack of efficacy of the compounds on MK-801-induced escape latency during the acquisition trial, the application of the compounds and the training were extended in these animals by one day. 

In the retention trial, only weak activity of DETA NONOate on reference memory in MK-801-treated mice was observed. 

Assuming, the present results are accurate, they indicate that, among the tested substances DETA NONOate is the most potent to improve pharmacologically induced disruptions in reference memory. 

The implication of NO in learning and memory processes is well-documented, and the procognitive activity of several NO donors and NOS inhibitors has been demonstrated in experimental models of schizophrenia and/or AD. The majority of papers concern the activity of sodium nitroprusside (SNP), molsidomine (both NO donors), or NCX2057 (NO releasing derivative) in the novel object recognition test (NOR) and in the models of attentional deficits (pre-pulse inhibition) [24,25,26,27,28,29,30,31,32]. The activity of presently used NONOates in NOR was demonstrated in our latest studies [8], which additionally have shown that NONOates and NPLA poorly reversed scopolamine-induced deficits in the T-maze, while being slightly more active in MK-801-treated animals, which is contrary to the present results.

Available studies concerning the activity of NO donors or NOS inhibitors in spatial memory tasks are relatively scarce. A few papers indicated that L-NAME, an NOS inhibitor, disrupted radial water maze learning when given alone or reversed ketamine-induced deficits, depending on the dose [28,33], indicating that NO activation might be essential for water maze learning. Other researchers showed that s-nitrosoglutathione (GSNO), a nitrosothiol, and a sustained NO releaser ameliorated behavioural and neurochemical changes in an experimental model of sporadic Alzheimer‘s disease (sAD) in rats [34,35]. The studies of Wass et al. indicated that the administration of a non-selective inhibitor of nitric oxide synthases, l-NAME, alleviated PCP-induced changes in the MWM [36,37,38]. PCP is a non-competitive antagonist of the NMDA receptor.

When comparing our results with previous works, both the advantages and limitations of the current studies can be discussed. As an advantage, we can consider the fact that a wide range of compounds with different activities towards nitric oxide have been used, i.e., slow and fast NO releasers and a selective nNOS inhibitor. The compounds were administered at the same time, in the same manner, and to the same animal species, thus any fluctuations due to seasons or animal metabolism and reactivity were minimalised. The reference memory was measured 24 h after the last training, and the drug treatment was not influenced by any manipulation. Concomitantly, as in the majority of other studies, we have also met some limitations. Pharmacologically, transient induction of cognitive decline may be considered one of them. Although the models are widely used and generally accepted, transgenic mice or rats may better resemble not only the behaviour, but also the neurochemical characteristics of the disease. The use of a selective nNOS inhibitor may constitute both an advantage and a disadvantage, depending on your point of view. Here we assume that the endothelial pool of NO is not affected by the treatment with NPLA, but in the studies cited above, l-NAME antagonised both eNOS and nNOS, as the compound is a non-selective NOS inhibitor. One cannot precisely establish to what extent both NO pools are impacted. As far as we know, no selective eNOS inhibitors are available. 

Our comparative studies indicate that both NO releasers and NPLA may prevent the induction of spatial learning deficits by scopolamine but are less active towards MK-801-induced deficits. Moreover, the fast releaser DETA NONOate seemed to perform better than the slow releaser spermine NONOate, being more effective in scopolamine-treated animals and slightly reversing MK-801-induced deficits as well. Notably, both NONOates release nitric oxide by spontaneously dissociating in a pH dependent manner and are useful for reliable and controllable NO generation in solution. Why their activity depends on the test used and/or interaction with tool compounds is not clear at present. 

The Morris water maze test is thought to measure hippocampal-dependent spatial memory, movement control, and cognitive mapping [39,40,41]. Contrary to, e.g., the T-maze, which is more structured and only requires the mouse to make a binary decision (i.e., choose left or right), in MWM the animal needs to continually decide where to go [42]. Thus, the test is constrained by several limitations that do not necessarily constitute a cognitive factor. To some extent, variances in performance scores can be due to differences in thigmotaxis—the tendency of animals to stay close to the walls of the pool—different tendencies of mice to float passively in the water, and swimming speed. It is clearly seen on the heat maps that in the experimental groups treated with MK-801, the thigmotaxis was more evident than in scopolamine-treated mice (Figure 15). These factors can be considered a bias of the experiment and contribute to the large individual discrepancies within the group. Our analysis revealed that the administration of both compounds increased locomotion immediately after administration during training sessions but failed to alter general motor activity or exploratory behaviour. In the test session, no changes in swimming velocity were observed. These results were not shown as they are in line with the results already published [43,44]. 

Some hypotheses regarding the mechanisms of action of the compounds in the MWM can be put forward. Learning impairment in the MWM develops as a consequence of hippocampal dysfunction and long-term potentiation deficits [45,46,47,48,49]. It is well described that scopolamine induces hippocampal cholinergic dysfunction (decreased acetylcholine levels, increased acetylcholinesterase activity, and decreased density and affinity of muscarinic receptors) as well as mitochondrial dysfunction (including oxidative stress). Therefore, scopolamine primarily impairs learning and memory by altering the cholinergic, oxidative, and inflammatory balance in the brain [20,50,51]. Similar dysregulations of the redox state are attributed to the action of MK-801 [52,53], although there is relatively less data concerning this at present. However, the compound, by exerting a direct inhibitory action on NMDA receptors, may have an additional detrimental impact on NMDA-related processes, including long-term potentiation and cGPM synthesis [4]. It may be hypothesised that both NONOates and NPLA reverse behavioural anomalies in the scopolamine-treated mice via oxidative stress quenching and enhancing antioxidative enzyme activity, but this may not be sufficient to prevent NMDA-related dysfunctions, which are essential in MWM learning. A variety of studies confirm that NO acts as a double-edged sword being detrimental and beneficial depending on the conditions. Thus, NO-related compounds may act as balancing agents that maintain homeostasis within the system. 

## 5. Conclusions

The present results indicate that scopolamine-induced dysfunctions in hippocampal-dependent spatial learning and memory can be alleviated with NO-related agents. The question arises if this is plausible in humans as well. There is no clear answer to this at the moment.

As far as we know, no NO-related compounds with the inclination to improve cognitive function are approved as formal drugs. However, a precursor to nitric oxide, as diet supplements are available, and are usually composed of l-arginine and l-citrulline. Generally, they may find use in various types of hypertension, ischemic heart disease, heart failure, atherosclerosis (especially of the lower limbs), hypercholesterolemia, glaucoma, chronic kidney failure, diabetes, and the prevention of cardiovascular disease, including stroke [54]. In some of these disorders, cognitive decline develops and, to some extent, may be alleviated with NO booster supplementation. Moreover, despite these positive effects, L-arginine supplementation under some circumstances may increase the production of NO, peroxynitrite, and reactive oxygen species (due to impaired eNOS activity), inducing rather detrimental and not beneficial effects. For this reason, it should be used with caution, especially by the elderly. It most likely refers to the putative use of any NO-related agent in humans. 

## Figures and Tables

**Figure 1 brainsci-13-00410-f001:**
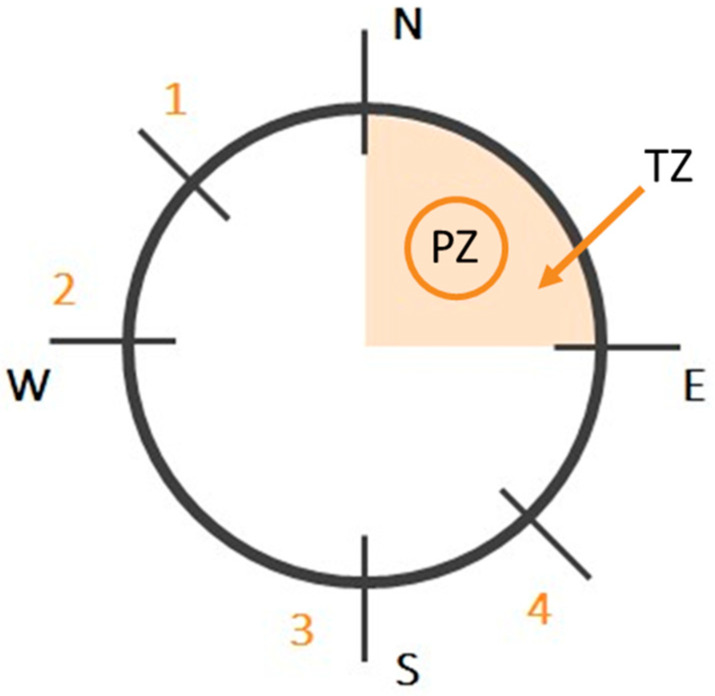
Schematic representation of the Morris water pool. PZ—platform; TZ—the area inside which the platform was located; 1, 2, 3, 4—starting points.

**Figure 2 brainsci-13-00410-f002:**
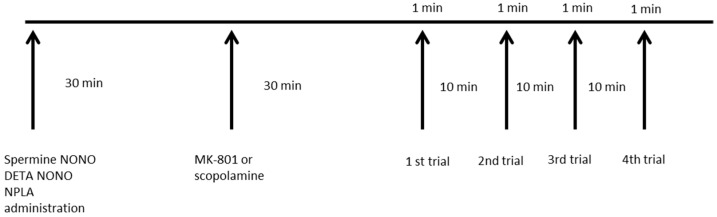
The daily chart of the experiment was repeated every day for 4 of 5 consecutive days (the acquisition phase). Control animals received vehicles in place of treatment.

**Figure 3 brainsci-13-00410-f003:**
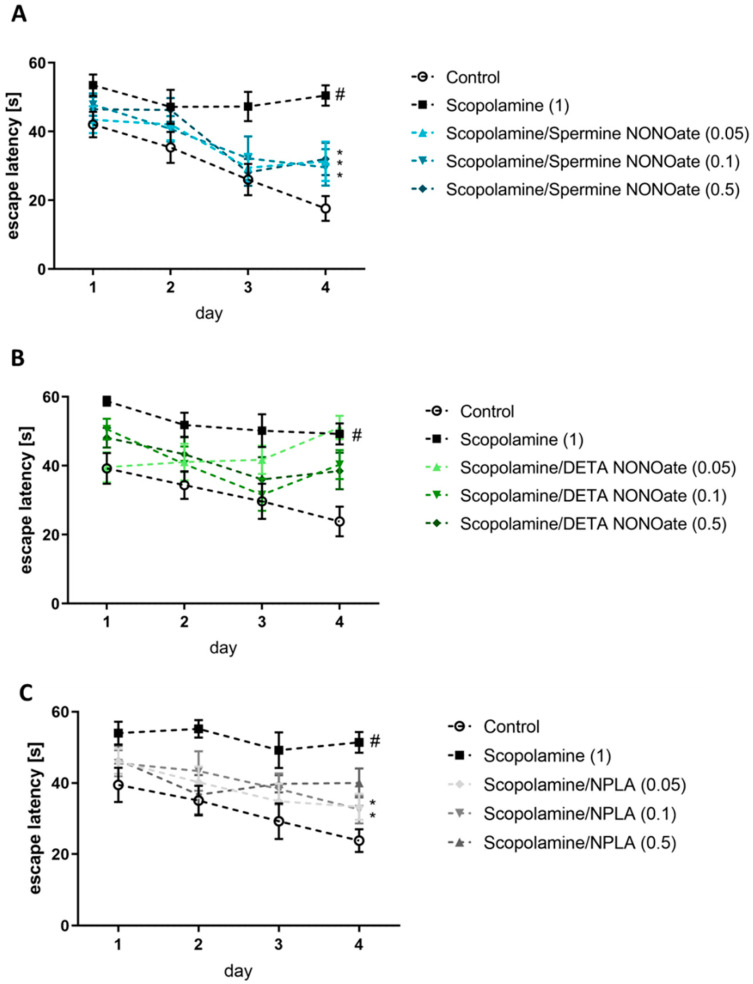
The effects of spermine NONOate (**A**), DETA NONOate (**B**), and NPLA (**C**) on scopolamine (1 mg/kg)–induced amnestic effect on the acquisition of spatial memory in the MWM of CD1 mice. The values are expressed as the means ± SEM. # *p* < 0.001 indicates the difference scopolamine-treated mice vs. the respective vehicle (N = 8–10), and * *p* < 0.05 indicates the difference between the scopolamine group and the treatment groups.

**Figure 4 brainsci-13-00410-f004:**
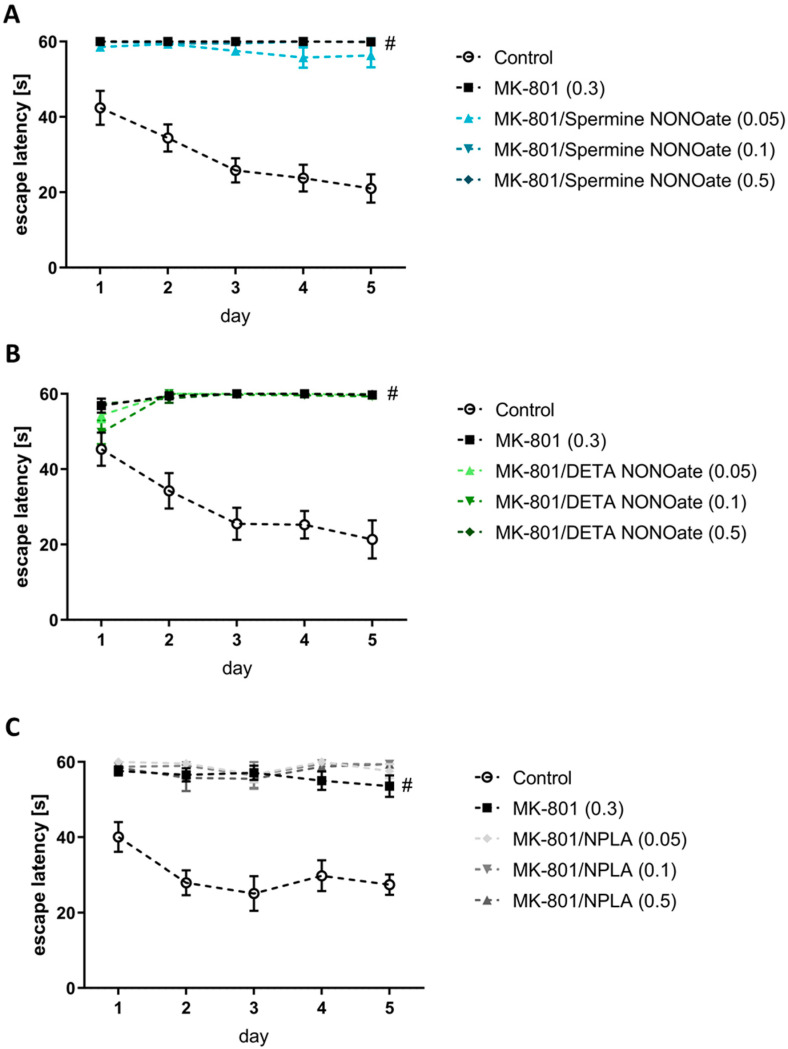
The effects of spermine NONOate (**A**), DETA NONOate (**B**), and NPLA (**C**) on the MK-801 (0.3)–induced amnestic effect on the acquisition of spatial memory in the MWM of CD1 mice. The values are expressed as the means ± SEM. # *p* < 0.001 indicates the difference between MK-801-treated mice vs. the respective vehicle (N = 8–10) (please consider that in Figure 4a lines, of MK-801/spermine NONOate (0.1 mg/kg) and MK-801/spermine NONOate (0.5 mg/kg) are not visible because they ideally overlap with the MK-801 line alone).

**Figure 5 brainsci-13-00410-f005:**
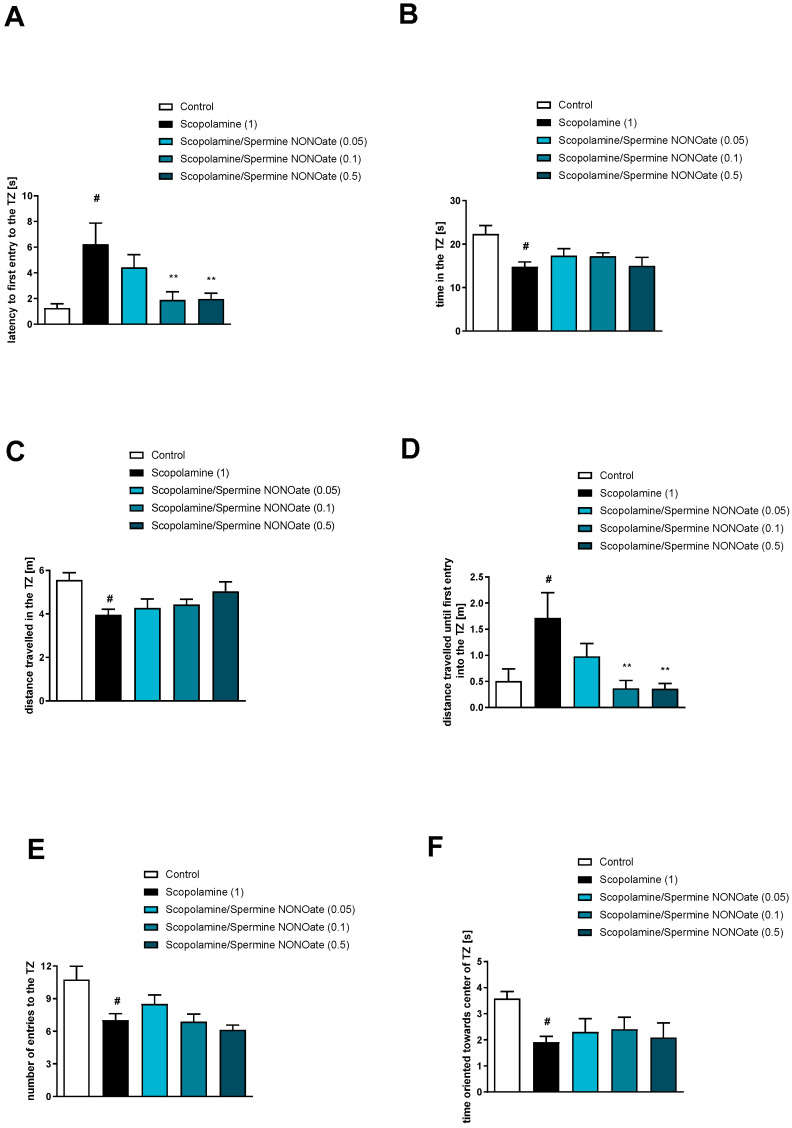
The effects of spermine NONOate on scopolamine-induced deficits in the retention of spatial memory in the MWM of CD1 mice. TZ-related parameters. (**A**) latency to the first entry into the TZ, (**B**) the time in the TZ, (**C**) the distance travelled in the TZ, (**D**) the distance travelled until the first entry into the TZ, (**E**) the number of entries into the TZ, and (**F**) the time oriented towards the centre of the TZ when inside the zone. The values are expressed as the means ± SEM. # *p* < 0.001 vs. control group and ** *p* < 0.005 vs. scopolamine-treated animals (N = 8–10).

**Figure 6 brainsci-13-00410-f006:**
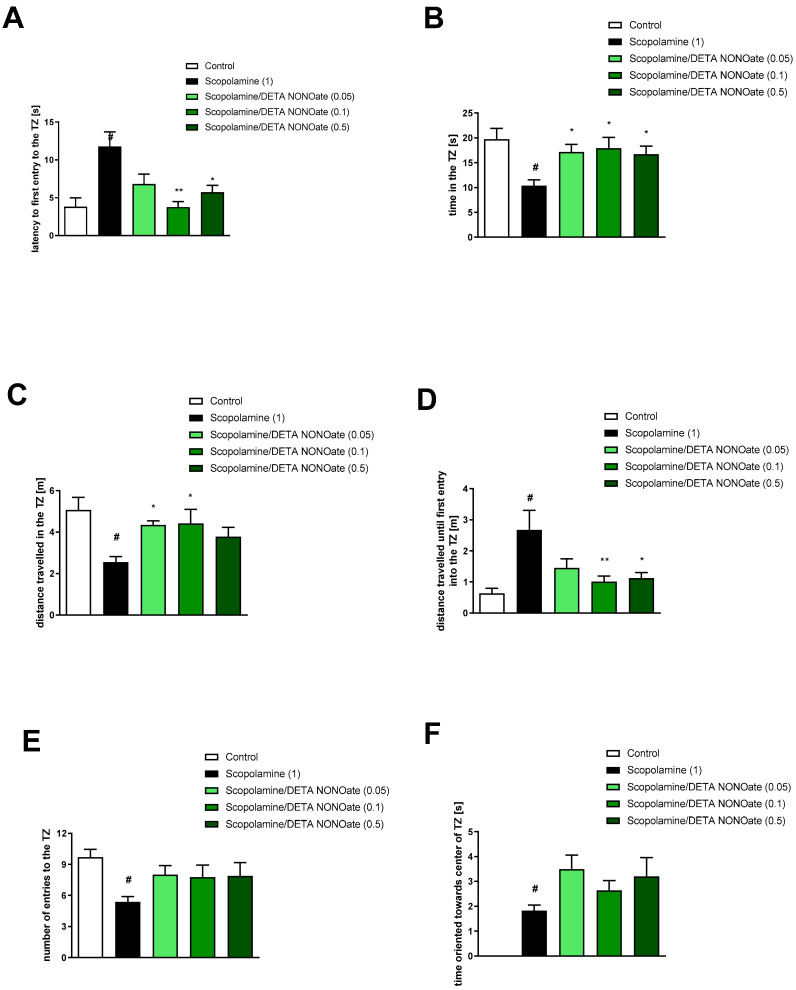
The effects of DETA NONOate on scopolamine-induced deficits in the retention of spatial memory in the MWM of CD1 mice. TZ-related parameters. (**A**) latency to the first entry into the TZ, (**B**) time in the TZ, (**C**) the distance travelled in the TZ, (**D**) the distance travelled until the first entry into the TZ, (**E**) the number of entries into the TZ, and (**F**) the time oriented towards the centre of the TZ when inside the zone. The values are expressed as means ± SEM. # *p* < 0.001 vs. control group and * *p* < 0.05; ** *p* < 0.005 vs. scopolamine-treated animals (N = 8–10).

**Figure 7 brainsci-13-00410-f007:**
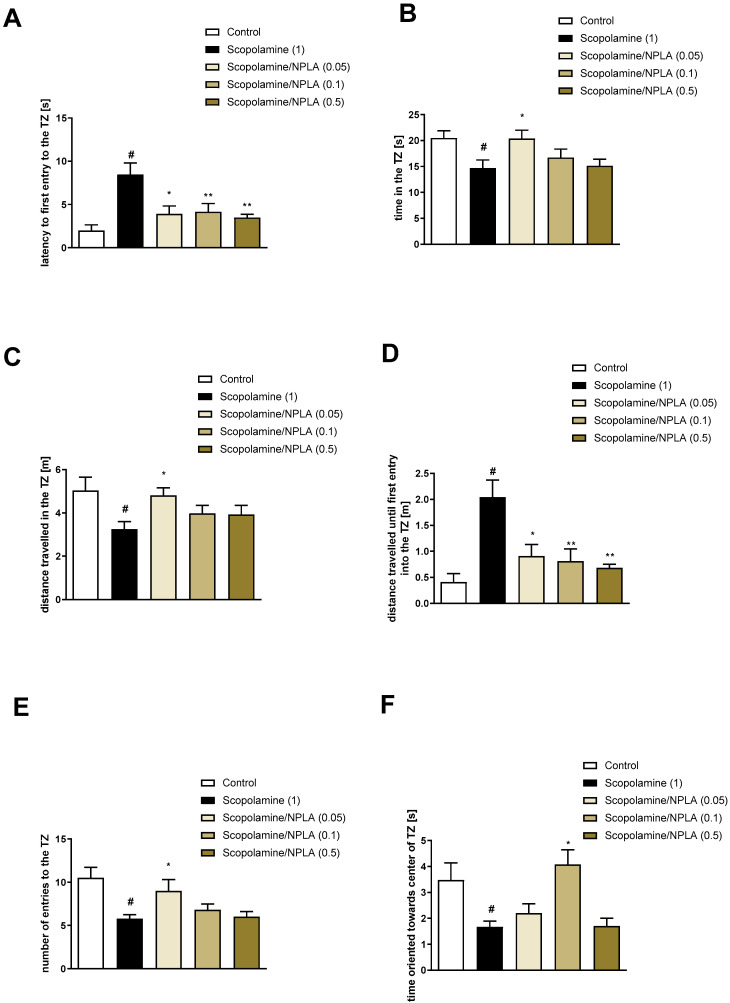
The effects of NPLA on scopolamine-induced deficits in the retention of spatial memory in the MWM of CD1 mice. TZ-related parameters. (**A**) the latency to the first entry into the TZ, (**B**) the time in the TZ, (**C**) the distance travelled in the TZ, (**D**) the distance travelled until the first entry into the TZ, (E) the number of entries into the TZ, and (**F**) the time oriented towards the centre of the TZ when inside the zone. The following values are expressed as the means ± SEM. # *p* < 0.001 vs. control group and * *p* < 0.05; ** *p* < 0.02 vs. scopolamine-treated animals (N = 8–10).

**Figure 8 brainsci-13-00410-f008:**
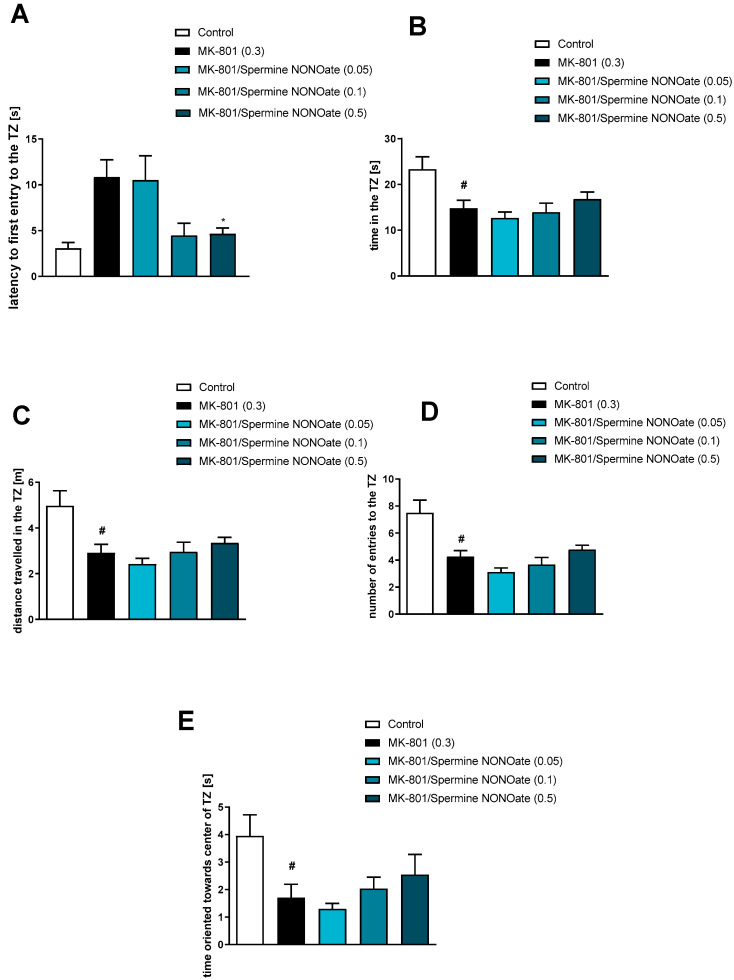
The effects of spermine NONOate on MK-801-induced deficits in the retention of spatial memory in the MWM of CD1 mice. TZ-related parameters. (**A**) the latency to the first entry into the TZ, (**B**) the time in the TZ, (**C**) the distance travelled in the TZ, (**D**) the number of entries into the TZ, and (**E**) the time oriented towards the centre of the TZ when inside the zone. The values are expressed as the means ± SEM. # *p* < 0.001 vs. control group and * *p* < 0.05 vs. scopolamine-treated animals (N = 8–10).

**Figure 9 brainsci-13-00410-f009:**
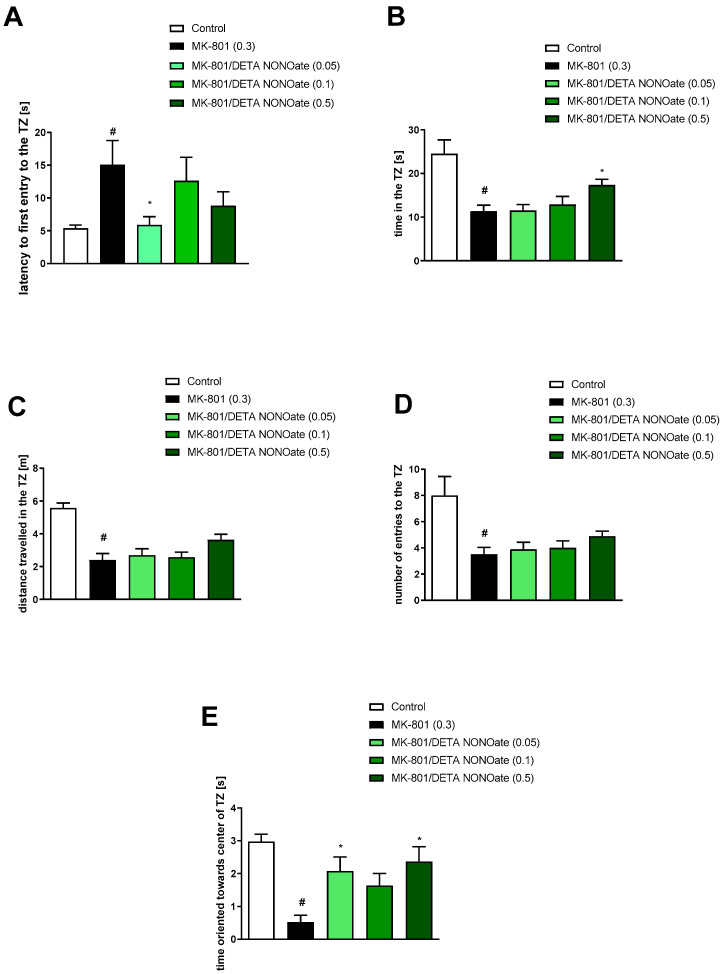
The effects of DETA NONOate on MK-801-induced deficits in the retention of spatial memory in the MWM of CD1 mice. TZ-related parameters. (**A**) the latency to the first entry into the TZ, (**B**) the time in the TZ, (**C**) the distance travelled in the TZ, (**D**) the number of entries into the TZ, and (**E**) the time oriented towards the centre of the TZ when inside the zone. The values are expressed as the means ± SEM. # *p* < 0.001 vs. control group and * *p* < 0.05 vs. scopolamine-treated animals (N = 8–10).

**Figure 10 brainsci-13-00410-f010:**
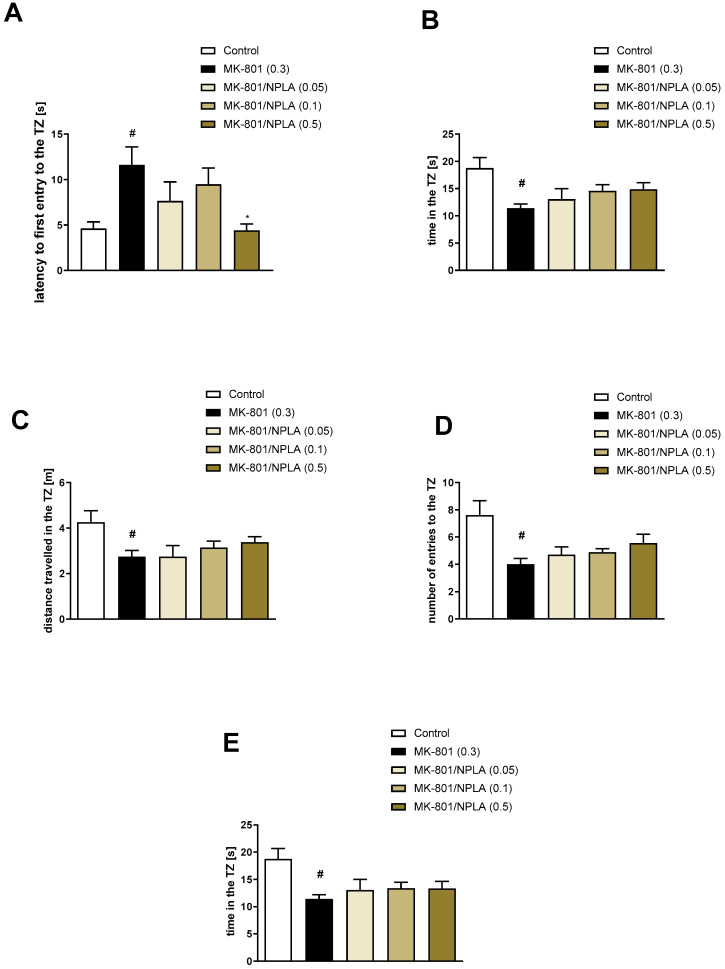
The effects of NPLA on MK-801-induced deficits in the retention of spatial memory in the MWM of CD1 mice. TZ-related parameters. (**A**) the latency to the first entry into the TZ, (**B**) the time in the TZ, (**C**) the distance travelled in the TZ, (**D**) the number of entries into the TZ, and (**E**) the time oriented towards the centre of the TZ when inside the zone. The values are expressed as the means ± SEM. # *p* < 0.001 vs. control group and * *p* < 0.05 vs. scopolamine-treated animals (N = 7–9).

**Figure 11 brainsci-13-00410-f011:**
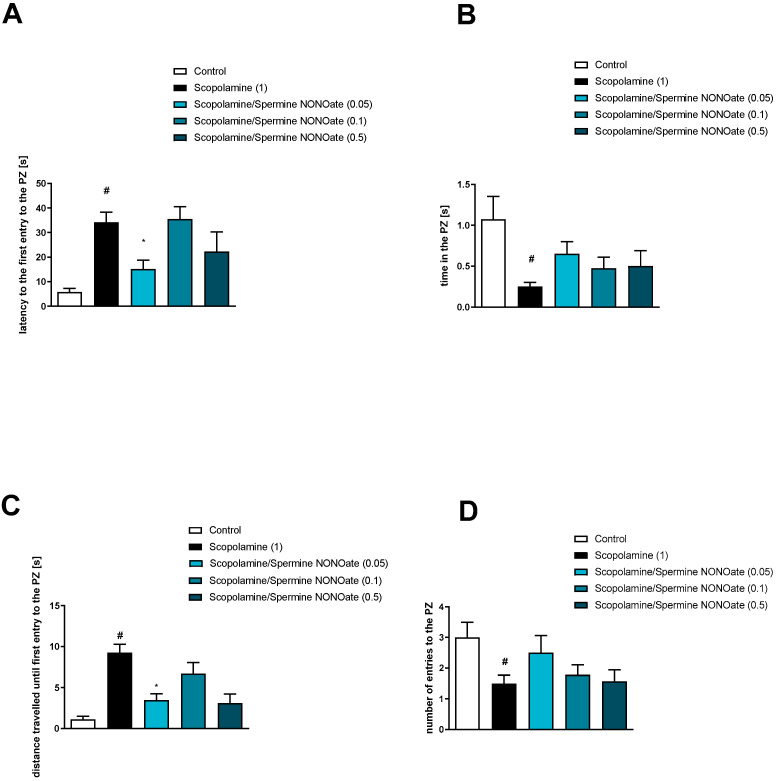
The effects of spermine NONOate on scopolamine-induced deficits in the retention of spatial memory in the MWM of CD1 mice. PZ-related parameters. (**A**) the latency to the first entry to the PZ, (**B**) the time in the PZ, (**C**) the distance travelled in the PZ, (**D**) the number of entries into the PZ. The following values are expressed as means ± SEM. # *p* < 0.001 vs. control group * *p* < 0.05 vs. scopolamine-treated animals (N = 8–10).

**Figure 12 brainsci-13-00410-f012:**
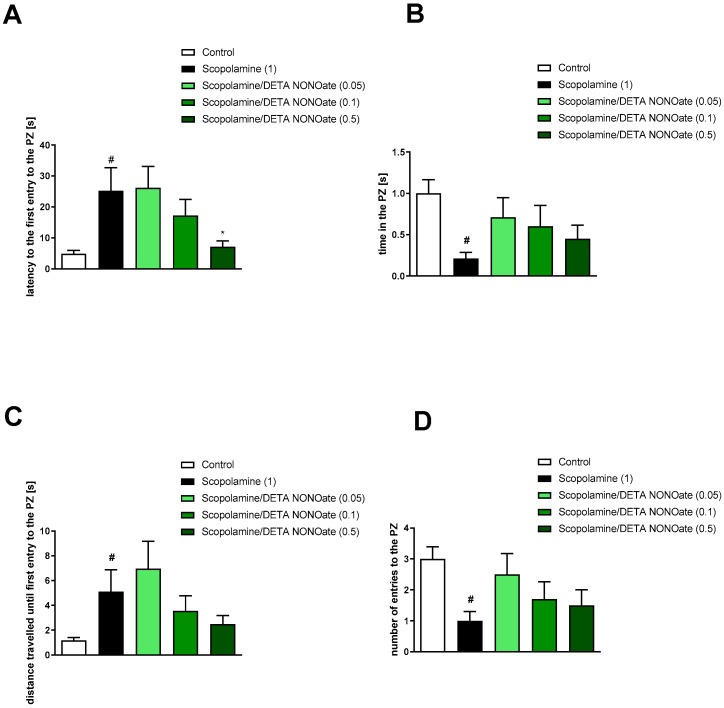
The effects of DETA NONOate on scopolamine-induced deficits in the retention of spatial memory in the MWM of CD1 mice. PZ-related parameters. (**A**) the latency to the first entry into the PZ, (**B**) the time in the PZ, (**C**) the distance travelled in the PZ, (**D**) the number of entries into the PZ. The values are expressed as the means ± SEM. # *p* < 0.001 vs. control group and * *p* < 0.05 vs. scopolamine-treated animals (N = 8–10).

**Figure 13 brainsci-13-00410-f013:**
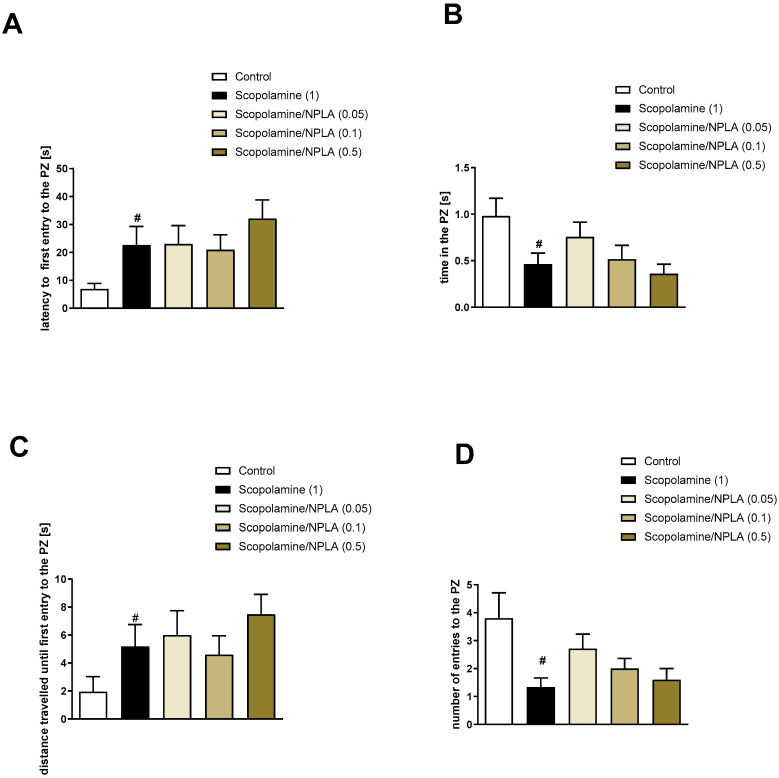
The effects of NPLA on scopolamine-induced deficits in the retention of spatial memory in the MWM of CD1 mice. PZ-related parameters. (**A**) the latency to the first entry into the PZ, (**B**) the time in the PZ, (**C**) the distance travelled in the PZ, (**D**) the number of entries into the PZ. The values are expressed as the means ± SEM. # *p* < 0.001 vs. control group (N = 7–9).

**Figure 14 brainsci-13-00410-f014:**
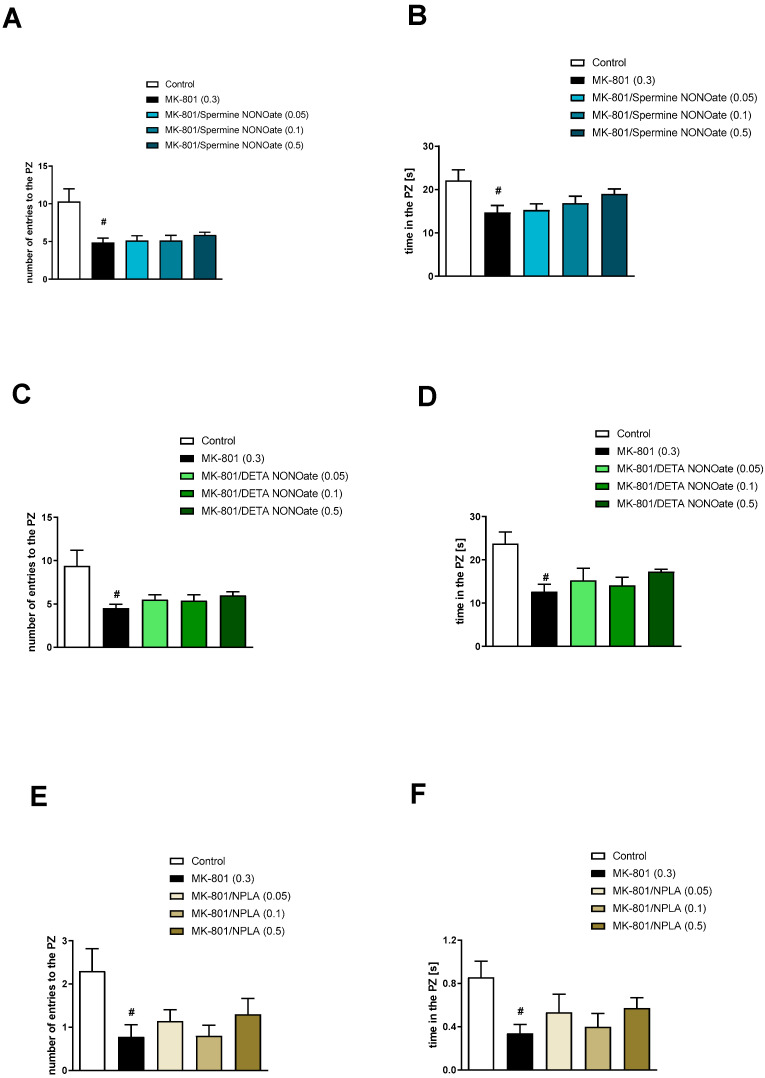
The effects of spermine NONOate (**A**,**B**), DETA NONOate (**C**,**D**), and NPLA (**E**,**F**) on MK-801-induced deficits in the retention of spatial memory in the MWM of CD1 mice. PZ-related parameters. The values are expressed as the means ± SEM. # *p* < 0.001 vs. control group (N = 8–10).

**Figure 15 brainsci-13-00410-f015:**
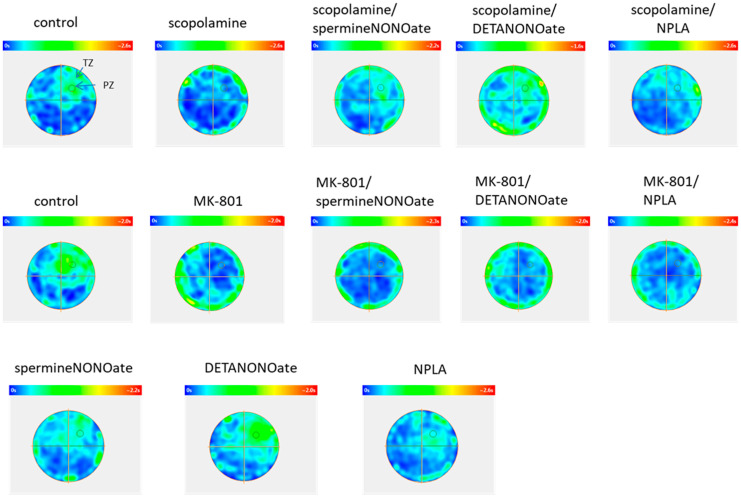
Heat maps reflecting the time spent swimming in particular areas of the maze. The maps show mean data for different treatment groups. Dark blue—no time spent in the area, red—the most time spent in the area. TZ—target zone; PZ—platform zone.

**Table 1 brainsci-13-00410-t001:** The activity of investigated compounds given alone in the MWM. TZ—target zone; PZ—platform zone. The results are presented as means ± SEM. The compounds were administered at a dose of 0.5 mg/kg (N = 8).

	Compound	Control	Spermine NONOate	DETA NONOate	NPLA
Parameter					
Latency to first entry into the TZ		2.25 ± 0.33	2.02 ± 0.33	5.9 ± 2.0	4.64 ± 2.49
Time in the TZ		20.49 ± 1.4	17.35 ± 1.74	24.89 ± 2.69	19.74 ± 1.9
Distance travelled in the TZ		5.5 ± 0.33	4.5 ± 0.34	5.19 ± 0.55	4.53 ± 0.39
Distance travelled until first entry into the TZ		0.5 ± 0.23	1.02 ± 0.33	0.78 ± 0.29	0.83 ± 0.36
Number of entries into the TZ		10.75 ± 1.2	8.4 ± 1.11	7.1 ± 0.87	1.22 ± 0.9
Time oriented towards the centre of the TZ		2.1 ± 0.65	3.1 ± 0.54	2.79 ± 0.43	0.5 ± 0.42
Latency to first entry into the PZ		6.9 ± 1.98	8.45 ± 1.2	9.8 ± 1.3	12.21 ± 5.62
Time in the PZ		1.07 ± 0.72	0.72 ± 0.26	0.87 ± 0.15	0.61 ± 0.16
Number of entries into the PZ		3.0 ± 0.49	2.4 ± 0.71	2.9 ± 0.5	2.33 ± 0.4

## Data Availability

Data supporting reported results can be purchased on request.

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
