# Peer review of "A Comparative Study of the Impact of NO-Related Agents on MK-801- or Scopolamine-Induced Cognitive Impairments in the Morris Water Maze"

_brainsci, 2023, doi:10.3390/brainsci13030410_

Round 1

Reviewer 1 Report

Abstract

- Given that there is a limit of 200 words based on the Journal's instructions, authors should try to shorten abstract's length.  For example they could omit the detailed information about MWM (l. 12-15 & 20-24). This could be included in the main text.

-l. 19: MWM also tests spatial learning (acquisition phase), not only memory.

-Keywords: Authors could use some words that do not also appear in the title.

Introduction

-l.37: endothelial NOS instead of eNOS since the sentence starts with this word. Same comment applies to nNOS (l. 43). Please make appropriate changes wherever necessary in the text.

-l. 42: what "i.a" stands for?

-l. 68: MWM is not an animal model of spatial learning. Do authors mean "a reliable and valid test of spatial learning and memory"? A reference should also be added. Same comment applies to AD and schizophrenia (l.70).

-Authors should try to state a hypothesis regarding the effect of NO-related agents on deficits induced by MK-801 and scopolamine.

Methods

-The groups (control/experimental) according to the manipulation received (type of drug, dose) and their size (number of mice/group) have to be reported. 

-Behavioral testing time should be mentioned (light/dark cycle?)

-Authors should clarify that time to reach the platform in acquisition trial is a measurement of spatial learning.

-l. 98: Please explain the reason the acquisition phase of scopolamine consisted of 4 days while of MK-801 5 days.

-l. 99: The inter-trial interval should be mentioned.

-l. 105-106: If understood well, the starting point for retention trial was #2, used also during acquisition trials. Is this correct? If so, there is a methodological issue since based on literature animal in retention phase should be placed into the pool from a novel position (180◦from the platform position during the acquisition phase).

-Given that in all training trials drugs were administered, it is not clear to the reviewer why this was not the case in acquisition trial as well.

-2.4: Authors analyzed many parameters in the retention trial and this is a little confusing. Is there a need to analyze so many measurements? Are they all considered reliable measurements of spatial retention or some indicate different behaviors? This should be clarified. Could you provide any references?

-2.5: Authors should clearly state which variables were analyzed by the specific statistical tests.

Results

-l. 147-148, 154-155: Even if differences are not statistically significant, authors should report statistical values.

-l. 154-155: To which effect do authors refer to? Not clear.

-Fig. 4: Not all lines representing the five groups are visible in figures A, B, C. For example, in A only three lines can be seen.

Discussion

-l. 309: "impaired learning deficits": Do authors mean "impaired spatial learning"?

- Some parts are mainly a repetition of the results (e.g., l. 310-342). Authors need to discuss their findings taking into account existing evidence.

Author Response

We thank the Reviewer for the valuable comments and remarks. We corrected the manuscript accordingly.

Abstract

- Given that there is a limit of 200 words based on the Journal's instructions, authors should try to shorten abstract's length.  For example they could omit the detailed information about MWM (l. 12-15 & 20-24). This could be included in the main text.

According to the Reviewer’s suggestion the Abstract was shortened

-l. 19: MWM also tests spatial learning (acquisition phase), not only memory.

corrected

-Keywords: Authors could use some words that do not also appear in the title.

Please see the key words

Introduction

-l.37: endothelial NOS instead of eNOS since the sentence starts with this word. Same comment applies to nNOS (l. 43). Please make appropriate changes wherever necessary in the text.

Corrected

-l. 42: what "i.a" stands for?

i.a (inter alia) was changed to more common e.g

-l. 68: MWM is not an animal model of spatial learning. Do authors mean "a reliable and valid test of spatial learning and memory"? A reference should also be added. Same comment applies to AD and schizophrenia (l.70).

Corrected according to Reviewer’s suggestion.

-Authors should try to state a hypothesis regarding the effect of NO-related agents on deficits induced by MK-801 and scopolamine.

Please see the text at the end of the Introduction.

NMDA or muscarinic receptors are expressed on neurons, endothelial cells and astrocytes, which are closely related to each other and create the minimal functioning unit in the brain called neurovascular unit (NVU) (…). Nitric oxide-dependent pathways are essential regulators of mutual interactions between cells and the fluctuations of NO-dependent activities (such as cGMP synthesis or the production of oxidative stress) translate into NVU dysfunctions that may contribute to the amnestic effect of MK-801 or scopolamine, and at least partially resemble these observed in schizophrenic or AD patients. We thus hypothesize that the administration of NO-related agents may counteract the MK-801 or scopolamine-induced dysfunctions by restoring brain homeostasis.

And also in discussion, lines: 439-455

Some hypotheses regarding the mechanisms of action of the compounds in the MWM can be put forward. Learning impairment in the MWM develops as a consequence of hippocampal dysfunction and long-term potentiation deficits [45-49]. It is well described that scopolamine induces hippocampal cholinergic dysfunction (decrease in acetylcholine level, increase in acetylcholinesterase activity, and decrease in density and affinity of muscarinic receptors), and mitochondrial dysfunction (including oxidative stress). Thus, scopolamine impairs the ability of learning and memory predominantly by changes of cholinergic, oxidative, inflammatory balance in the brain [50, 51]. Similar dysregulations of redox state is attributed to the action of MK-801 [52, 53], although there is relatively less data concerning this at present. However, the compound, by direct inhibitory action on NMDA receptors, may have additional detrimental impact on the NMDA-related processes, including long-term potentiation and cGPM synthesis [4]. It may be hypothesized that both NONOates and NPLA reverse behavioural anomalies in the scopolamine treated mice via oxidative stress quenching and enhancing antioxidative enzyme activity, but this may be not sufficient to prevent NMDA-related dysfunctions which are essential in MWM learning. Variety of studies confirm that NO acts as a double-edged sword, being detrimental and beneficial depending on conditions. Thus NO-related compounds may act as balancing agents that maintain the homeostasis within the system.

Methods

-The groups (control/experimental) according to the manipulation received (type of drug, dose) and their size (number of mice/group) have to be reported. 

We added some sentences regarding this in M&M section. The schedule of the administration is indicated on Fig. 2. Also everything is included in the descriptions under each graph (treatments, number of animals in groups).

-Behavioral testing time should be mentioned (light/dark cycle?)

Corrected

Line:

-Authors should clarify that time to reach the platform in acquisition trial is a measurement of spatial learning.

Corrected, line

-l. 98: Please explain the reason the acquisition phase of scopolamine consisted of 4 days while of MK-801 5 days.

We included some sentences regarding this in M&M and Discussion section

Please see lines :

-l. 99: The inter-trial interval should be mentioned.

It was several minutes; we include this information in the text

Lines:

-l. 105-106: If understood well, the starting point for retention trial was #2, used also during acquisition trials. Is this correct? If so, there is a methodological issue since based on literature animal in retention phase should be placed into the pool from a novel position (180◦from the platform position during the acquisition phase).

The Reviewer is right and thank for this remark. We corrected this in the manuscript, line .

 Mice were released from the WS start point and were allowed to swim freely for 60 s.

However, in most papers the starting location is not clearly indicated.

-Given that in all training trials drugs were administered, it is not clear to the reviewer why this was not the case in acquisition trial as well.

Did the Reviewer mentioned retention trial ?

We reviewed many papers concerning MWM as well as we did many test experiments while setting up the method.

The details concerning the performing of MWM differ between laboratories. There is no single rigid scheme for performing this test, both in terms of the number of training days, the number of training sessions per day, and the administration scheme. Just different protocols are acceptable, plausible and credible (Vorhees and Williams, 10.1038/nprot.2006.116).

Several characteristics have contributed to the prevalent use of the MWM. These include the lack of required pretraining, its high reliability across a wide range of tank configurations and testing procedures, its cross-species utility (rats, mice and humans (in a virtual maze)), extensive evidence of its validity as a measure of hippocampally dependent spatial navigation and reference memory5, its specificity as a measure of place learning, and its relative immunity to motivational differences across a range of experimental treatment effects (Vorhees and Williams, 10.1038/nprot.2006.116).

The aim of our experiments was to investigate the spatial memory (or reference memory) 24 hours after the last administration of the compounds. Such an approach allows to measure the reference memory when animal’s behaviour is not influenced by the compounds. Similar protocol is used by e.g Sałat et al. 2015, 2017). Also in many protocols no clear indication of the last drug administration before the probe test is given (Wass et al. 2006 a,b, 2008;  doi: 10.1016/j.pbb.2008.01.011 ·  DOI: 10.1016/j.bbr.2006.07.003  ·  DOI: 10.1016/j.bbr.2006.03.036 ).

To assess reference memory at the end of learning, a probe (transfer) trial is given. The most common method is to administer one probe trial 24 h after the last acquisition day. With some procedures, the probe trial is administered immediately following the last learning trial; however, this cannot differentiate between short- and long-term memory, as it may reflect memory for the most recent training session. A long interval between the last training trial and the probe trial is essential if reference memory is to be determined independent of the memory of the last training session. In some studies even several days interval is used.

-2.4: Authors analyzed many parameters in the retention trial and this is a little confusing. Is there a need to analyze so many measurements? Are they all considered reliable measurements of spatial retention or some indicate different behaviors? This should be clarified. Could you provide any references?

Many variations can be added to the basic MWM measures (which is latency to reach the TZ) and these can add valuable information for understanding the deficits that are observed or may even unmask more subtle deficits that are not seen. We include the measurements concerning TZ (the whole quadrant in which the platform was located) and also the measurements concerning the exact location of the platform, which is a small place of 10 cm in diameter and requires the precision from the animal in finding this area. Maei et al. 2009 Reviewed many papers up to 2009 showing that different parameters could be measured. In fact, the more accurate and refined tools are at the experimenter's disposal, the more parameters can be measured during the test probe. In the present studies we presented those parameters that were  affected by the tool compounds (scopolamine and MK-801), but we could analyze many more, that were not affected. Thus the study brings the information regarded not only well-known parameter, such escape latency or latency to the first entry, which are most commonly shown, but also the other parameters that may be informative (the more parameters are reversed by the compounds the more effective it is). 

-2.5: Authors should clearly state which variables were analyzed by the specific statistical tests.

We stated it clearly in this section, please see

  1. a) repeated-measures ANOVA was used to evaluate escape latency (the measurement of spatial learning) between groups in the training days during acquisition phase
  2. b) Student’s t-test was used to compare controls vs. MK-801 or controls vs scopolamine-treated animals (was applied to all tested parameters in the retention phase)
  3. c) one-way ANOVA followed by Dunnett’s post hoc comparison was used to evaluate the activity of the compounds in the retention phase (was applied to all tested parameters). Treatment groups receiving drugs+MK-801 were compared with MK-801 group and treatment groups receiving drugs+scopolamine were compared with scopolamine group.

Results

-l. 147-148, 154-155: Even if differences are not statistically significant, authors should report statistical values.

Values have been included, please see highlights in appropriate parts of the manuscript

-l. 154-155: To which effect do authors refer to? Not clear.

Corrected. Lines

When administered alone the investigated compounds had no impact on escape latency during the acquisition phase. The effect of spermineNONOate, DETANONOate and NPLA (administered at the dose of 0.5 mg/kg) on spatial learning was comparable to controls (data not shown).

-Fig. 4: Not all lines representing the five groups are visible in figures A, B, C. For example, in A only three lines can be seen.

This is because the lines are almost identical and overlap each other. We mentioned it in the figure legend.

Discussion

-l. 309: "impaired learning deficits": Do authors mean "impaired spatial learning"?

Yes. We corrected it in the text

- Some parts are mainly a repetition of the results (e.g., l. 310-342). Authors need to discuss their findings taking into account existing evidence.

We reorganized the discussion, also taking into consideration the comments of the other Reviewer. The repetition of the results are removed now.

Reviewer 2 Report

Well-designed work.

1. Please provide a detailed exploration and comparison with the previous similar work addressing the advantages and limitations of current results.

2. Please give an opinion form a translational perspective or evidence from clinical trials.

Technical points: lines from 65 to 79 move to Methodology. In line 362, in the discussion, remove underlined text.

Author Response

We thank the Reviewer for such a positive opinion and also valuable comments that improved the manuscript  considerably.

Well-designed work.

  1. Please provide a detailed exploration and comparison with the previous similar work addressing the advantages and limitations of current results.

We did accordingly; please see text lines: 376-413

  1. Please give an opinion form a translational perspective or evidence from clinical trials.

We did accordingly; please see lines: 463-475 in Conclusions.

Technical points: lines from 65 to 79 move to Methodology. In line 362, in the discussion, remove underlined text.

corrected

Round 2

Reviewer 1 Report

Authors have addressed all points raised by the reviewer.